# The Association between the Type of Delivery and Factors Associated with Exclusive Breastfeeding Practice among Polish Women—A Cross-Sectional Study

**DOI:** 10.3390/ijerph182010987

**Published:** 2021-10-19

**Authors:** Julia Tracz, Danuta Gajewska, Joanna Myszkowska-Ryciak

**Affiliations:** Department of Dietetics, Institute of Human Nutrition Sciences, Warsaw University of Life Sciences (WULS), 159C Nowoursynowska Str., 02-776 Warsaw, Poland; joanna_myszkowska_ryciak@sggw.edu.pl

**Keywords:** exclusive breastfeeding, breastfeeding duration, caesarean delivery, vaginal delivery

## Abstract

The type of delivery influences breastfeeding, both in terms of initiation and duration. The aim of the study was to determine the association between the type of delivery and factors associated with exclusive breastfeeding (EBF) practice among Polish women. Data on sociodemographic variables, pre-pregnancy weight, height, course of pregnancy, type of delivery and duration of breastfeeding were collected using a Computer-Assisted Web Interview. Of the 1024 breastfeeding women who participated in the study, 59.9% gave birth vaginally and 40.1% gave birth by caesarean section. The chance of starting EBF [OR: 0.478; 95% Cl: 0.274, 0.832] and continuing it for four months [OR: 0.836; 95% Cl: 0.569, 0.949] was lower in the case of caesarean delivery. Starting EBF was negatively affected by pre-pregnancy overweight status and obesity in the case of caesarean delivery. EBF practice for four months was negatively affected by age [18–24 years and 25–34 years], elementary education and average income [2001–4000 PLN] in the case of caesarean delivery. A negative impact on the chance of EBF for six months was also observed for younger age [18–24 years], elementary and secondary education and average income [2001–4000 PLN] in the case of caesarean delivery. There was no association between starting EBF and age, net income, place of living, pregnancy complications or the child′s birth weight category in the case of both subgroups, as well as between education and previous pregnancies in the case of vaginal delivery. These results suggest that women who deliver by caesarean section need additional breastfeeding support.

## 1. Introduction

Breastfeeding has many proven benefits for both children and mothers. Its protective effect against diarrhoea and respiratory system infection among children is well documented [1]. Breastfeeding reduces the risk of necrotizing enterocolitis and sudden infant death syndrome, as well as the risk of being overweight and obese in adult life [2,3]. Furthermore, breastfeeding is positively associated with a child’s intelligence quotient (IQ) [3]. A recent study showed that individuals who breastfed as children were less likely to contract COVID-19. As regards women’s health, breastfeeding decreases postpartum blood loss and speeds up involution of the uterus. Breastfeeding promotes a faster return to the pre-pregnancy body mass index [2], and there is an association between breastfeeding and reduced maternal depression [4]. Long-term effects for the mother, such as a lower risk of breast and ovarian cancers, are also proven [2,4]. Breastfeeding women, especially those who exclusively breastfeed, have longer periods of amenorrhoea [4]. It is worth emphasising that breastfeeding is also beneficial to the economy and the environment [5].

The World Health Organization (WHO) and The American Academy of Pediatrics (AAP) consentaneously recommend that children should be exclusively breastfed for the first six months of life [2,6]. The Polish Society for Paediatrics Gastroenterology, Hepatology and Nutrition also highlights that exclusive breastfeeding (EBF) for six months should be the paramount goal to achieve. In addition, a partial or shorter breastfeeding duration is also beneficial and should be continued as long as it is desired by the mother and the child [7]. Nevertheless, the factual duration of EBF or any breastfeeding is often very far from the recommended standards. The study by Królak-Olejnik et al. showed that the breastfeeding initiation rate in Poland was 97%, whereas the EBF rate at six months was only 4% [8].

Previous studies have reported that caesarean section is negatively associated with breastfeeding practice [9,10]. A prospective cohort study found that caesarean delivery has a negative impact on EBF during the first and third months after delivery but not during the sixth month [11]. More attention should be paid to early breastfeeding after caesarean section to avoid long-term health problems [12]. Unfortunately, an increase in the number of caesarean sections is noticed in many countries, with a rapid expansion in Eastern Europe, Central and South Asia [12]. In Poland, caesarean deliveries accounted for 37.25% of all deliveries in 2019 [13]. According to the WHO’s statement, caesarean sections are beneficial for saving maternal and infant lives but only when medical reasons occur. However, even when more than 10% of deliveries in a population are by caesarean section, there is still no impact on maternal and newborn mortality [14].

Therefore, factors which can increase the odds of starting EBF should be taken into account during antenatal care. Taking relevant actions among women who deliver the baby by caesarean section might enable them to breastfeed longer. The aim of this study was to determine the association between the type of delivery and factors associated with EBF practice among Polish women.

## 2. Materials and Methods

### 2.1. General Information

Survey data were obtained from 1024 mothers who had breastfed or had been breastfeeding at the time of the data gathering. Data collection was carried out in April 2018 and in May 2018. This study was a part of the educational program for parents, *Nestlé Healthy start to the future*, which was coordinated by the Polish Society of Dietetics with the participation of the IQS Agency (a Polish research-analytics company, Warsaw, Poland). A detailed research methodology has been described in previously published articles on dietary habits and nutritional status of women during pregnancy and breastfeeding [15] and factors influencing the duration of breastfeeding [16]. The study was conducted in accordance with the Declaration of Helsinki.

### 2.2. Study Participants and Collecting Data

Eligibility criteria were: being an adult (≥18 years old), a breastfeeding woman and being a mother of an infant or a toddler aged 6–18 months. Exclusion criteria included being pregnant, being a woman who has never breastfed, being the mother of an infant aged <6 months or a toddler aged over 18 months.

The study was conducted using the Computer-Assisted Web Interview (CAWI). The tool used to carry out the research was a questionnaire. Respondents were asked to complete the individual electronic form on the web panel (Opinie.pl) belonging to the independent external research agency, IQS. This web panel has existed since 2007. Currently, it has 110,000 active users including 10,500 mothers of children born between 2015–2018. Participation in the study was anonymous and voluntary, and women were not compensated for the participation. The research company provided the sampling frame and the IT tools needed for the fieldwork. The time for filling out the questionnaire did not exceed 15 min.

Sociodemographic data such as the woman’s age (open-ended question), education, number of persons in the household, place of living and net household income were collected.

Some socio-demographic variables were re-categorized prior to analyses. Age was re-categorized into three categories: “18–24 years”; “25–34 years”; and “over 35 years”. Education level was re-categorized into three categories: “elementary”, “secondary” and “tertiary”. Place of living was re-categorized into three categories based on the divisions presented in basic urban statistics from 2016: “rural area/farm”; “town < 99,000 inhabitants”; and “city with over 100,000 inhabitants” [17]. Net household income was categorized in line with a minimum salary and an average wage in Poland in 2018 into three categories: “≤2000 PLN”; “2001–4000 PLN”; “≥4001 PLN” [18].

Self-reported pre-pregnancy weight and height were used to calculate pre-pregnancy Body Mass Index (BMI). Pre-pregnancy BMI was calculated according to the following equation: person’s weight in kilograms divided by the square of the person’s height in meters. In accordance with the WHO classification, four BMI categories were defined: underweight (<18.5 kg/m^2^); normal body weight (18.5–24.9 kg/m^2^); overweight (25–29.9 kg/m^2^) and obesity (>30 kg/m^2^) [19]. Then, pre-pregnancy BMI was re-categorized into four groups: “underweight”, “normal body weight”, “overweight/obesity” and “no data”.

In the question on the course of pregnancy, women chose between “proper course of pregnancy” or “complicated course of pregnancy”. Respondents were asked whether it was their first pregnancy (“yes” or “no”) and whether their child was born “in due date”, “as a premature baby” or “after due date”. Women were asked to provide the birth weight of their child (open-ended question). The birth weight of the child was categorized into three groups: “low birth weight”—<2500 g, “normal birth weight”—2500–4000 g and “high birth weight”—>4000 g [20].

Participants were asked whether they had “a vaginal birth”, “a planned caesarean delivery” or “an unplanned caesarean delivery”. According to the type of delivery, all respondents were divided into two groups: “women with vaginal birth” and “women with caesarean delivery”.

Women were questioned about the duration of breastfeeding (“less than one month”; “one month”; “two months”; “three months”; “four months”; “five months”; “six months”; “seven months”; “eight months”; “nine months”; “ten months”; “twelve months” or “longer than one year”) and EBF (“I wasn’t exclusively breastfeeding”; “less than one month”; “one month”; “two months”; “three months”; “four months”; “five months” and “six months”. EBF was defined as feeding the infant with only breastmilk without any solids or liquids [21]. Duration of EBF was re-categorized into three categories: “starting EBF”; “EBF for four months”; “EBF for six months”.

### 2.3. Statistical Analysis

All data were analysed using STATISTICA version 13.1 (Copyright©StatSoft, Inc., 1984–2014, Cracow, Poland). Data were analysed in the groups of women with vaginal delivery (VD) and caesarean delivery (CD) in terms of age, level of education, place of living, net household income, pre-pregnancy body weight status, number of pregnancies, time of delivery, course of pregnancy and EBF duration. The normality of distribution of quantitative data was checked using the Shapiro–Wilk test. Variables not normally distributed were tested with the Mann–Whitney U test. Statistical significances for nominal (categorical) variables were determined using the Pearson’s chi-square test. An odds ratio (the Wald test) with 95% confidence intervals (95% CI) was calculated using logistic regression analysis to study the relationship between method of delivery and breastfeeding duration. The significance level for all statistical analyses was considered at *p* ≤ 0.05.

## 3. Results

### Characteristics of the Study Group

The total sample group consisted of 1024 women aged between 18 and 50 years old (average age was 30.6). Of all the participating women, 613 (59.9%) gave birth vaginally and 411 (40.1%) gave birth by caesarean section. There were no significant differences in the mean age (30.8 ± 7.4 vs. 30.3 ± 6.6; *p* = 0.552) and mean pre-pregnancy BMI (23.3 ± 6.0 vs. 23.8 ± 6.3; *p* = 0.200) between these two subgroups. The overall characteristics of the study sample are displayed in Table 1. All data are expressed as number values and in percentages.

There were no significant differences in the majority of the analysed variables—namely, age, education, place of living, net income, pre-pregnancy BMI and previous pregnancies—except for pregnancy complications. As was expected, the prevalence of pregnancy complications was higher in women who gave birth by caesarean section. Table 2 shows the relationship between the practice of EBF and the type of delivery. Significant differences were observed for the starting EBF and EBF for four months by the type of delivery. In the case of women giving birth by caesarean section, the chance of starting EBF was nearly half as high compared to women giving birth naturally. Similarly, the chance of continuing EBF for four months was lower by approximately 25% in the case of caesarean delivery. However, no differences were observed for the prevalence and odds ratio for EBF for six months between the two analysed subgroups.

Statistical analysis (the chi-squared test) provided more detailed information on the association between respondents’ characteristics, method of delivery and EBF practice (Table 3). Starting EBF was related to education, pre-pregnancy BMI and earlier pregnancies, but only in the case of women who gave birth by caesarean section. There was no association between starting EBF and age, net income, place of living, pregnancy complications or the child′s birth weight category in the case of both subgroups, as well as between education and previous pregnancies in the case of vaginal delivery. Continuation of EBF for four months was related to age, education and net income, both in the case of women giving birth vaginally and by caesarean section. Significant relationships between EBF and age and education were also observed in both subgroups at six months. 

For further statistical analyses (logistic regression), the variables that significantly differentiated the EBF practice in two subgroups were selected, namely: age, education, net income, pre-pregnancy BMI and previous pregnancies (Table 4). In the case of vaginal delivery, none of the examined variables influenced the odds ratio for EBF initiation. On the contrary, significant relationships were observed in the case of caesarean section: the chance of starting EBF was over three times higher among multiparous women and women with secondary education. However, excessive pre-pregnancy body weight reduced the odds ratio of breastfeeding to 34% in this subgroup. Overweight status or obesity before pregnancy also reduced the chance of continuing EBF for four months among women with vaginal delivery. Younger age (18–24 years old) significantly reduced the chance of EBF for four and six months, both in the case of vaginal delivery and caesarean section. In the case of the latter, women aged 25–34 also had about a 50% lower odds ratio of continuing EBF for four months. A negative impact on the chance of EBF for four and six months was also observed for elementary education in both subgroups. Moreover, the odds ratio for breastfeeding for six months was halved for women with a caesarean delivery and secondary education level. Although the level of income had no effect on the chance of starting EBF, some relationships were observed with the continuation of breastfeeding practice. The lowest net incomes nearly halved the odds ratio for four months of EBF in the case of natural delivery, while no such effect was observed for caesarean delivery. On the contrary, in the case of caesarean delivery, the average income category significantly reduced the chance of EBF for four and six months.

Figure 1 shows the percentage of women who declared breastfeeding, including EBF, during the first six months after labour depending on the type of delivery. Breastfeeding was more prevalent among women who had a vaginal delivery compared to women who had a caesarean section. The same relationship was observed in the case of EBF. It is worth noting that by the fourth month, the rate of decline in the percentage of breastfeeding and exclusively breastfeeding women was similar. However, after four months, a faster decline in the percentage of exclusively breastfeeding women was observed. Interestingly, this decrease was more pronounced in the case of vaginal delivery compared to the caesarean section delivery.

## 4. Discussion

EBF is the optimal way of feeding infants for six months of their lives. In this study, we investigated factors which can influence starting EBF and its duration in the case of caesarean section. Age, education, net income, pre-pregnancy BMI and previous pregnancies were significantly associated with EBF among women who delivered by caesarean section.

Our results showed that caesarean delivery had an impact on starting EBF and EBF at four months but not at six months after delivery. Caesarean delivery was negatively associated with the early EBF practice. Overall, these findings are in accordance with results reported by previous authors [11,22]. In the study by Hobbs et al., women who delivered by caesarean section were less likely to initiate breastfeeding and more likely to cease breastfeeding at four months postpartum [23]. A systematic review and meta-analysis of 53 studies concerning breastfeeding after caesarean section found that the rate of early breastfeeding after caesarean section was lower than after vaginal delivery [24]. However, it was reported that only the induced vaginal delivery and emergency caesarean section were associated with lower rates of breastfeeding at four weeks and at six months, respectively, which might have resulted from the better preparation taken by mothers who had a planned caesarean section [25]. In the present study, the proportion of caesarean sections was slightly higher (40.1%) than in previously published data (37.2%) [13]. High rates of caesarean sections should be an alarming issue, especially when breastfeeding practice is considered. Firstly, there is a need for familiarising women with benefits and risks of vaginal delivery as well as caesarean section. In Poland, women’s knowledge in this area is insufficient [26]. Secondly, health care professionals should provide mothers who deliver by caesarean section with additional breastfeeding support. This support should cover help from the midwife and the nurse, as well as physical and psychological support. In addition, dietary counselling should be provided to advise on an appropriate diet for breastfeeding women. This support should comply with those mothers’ special needs, which are caused by caesarean section.

In our study, women with secondary education who delivered by caesarean section were more likely to start EBF in this subgroup. However, we found that secondary education lowered the odds ratio of breastfeeding at six months, whereas Lio et al. found that women with medium education had higher chances of EBF for six months [27]. A difference between these results can be attributed to the fact that, in Poland, a long break at work among better educated women might worsen their position on the labour market [28]. Moreover, elementary education was associated with a lower chance of EBF both at four and six months, respectively. These results are consistent with previous studies wherein lower education increased the risk of early cessation of breastfeeding [29,30,31,32]. In the study by Logan et al., women with more than 12 years of education had increased odds of initiation and duration of breastfeeding. Early breastfeeding cessation was associated with a lower education level. Among mothers with a lower level of education, rates of smoking, overweight status and obesity, and elective caesarean section were higher [33]. Heck et al. suggested that both maternal and paternal education is correlated with starting breastfeeding. Parents with higher educational levels are more likely to comprehend benefits of breastfeeding [34]. Furthermore, our study indicated that a lower income decreased the chance of continuing EBF for four months in the case of vaginal delivery, whereas an average income decreased the odds ratio of EBF for four and six months in the case of caesarean section. However, Economou et al. found that a higher family income was a determinant of breastfeeding but not of EBF [35]. In another study, women with a higher socioeconomic status were less likely to stop EBF in the early postpartum period [36]. Intervention aimed at prolonging breastfeeding duration must consider the level of education of women regardless of the type of delivery. Furthermore, developing programmes directed to both parents could also be beneficial.

According to Simko et al., a higher risk of caesarean section was associated with pre-pregnancy obesity [37]. Obese women have double the risk of caesarean delivery [38]. Other authors pointed out that an overweight status and obesity were associated with complications such us as gestational diabetes, gestational hypertension, preeclampsia, preterm birth, caesarean delivery and macrosomia [39]. In our study, the odds ratios of starting breastfeeding were lower for those women with an overweight status or obesity who delivered by caesarean section. Contrary to our results, in the study by de Jersey et al., caesarean delivery in both women who were a normal weight and overweight was negatively associated with EBF at discharge from hospital [40]. However, in the Australian cohort, the risk of unsuccessful starting EBF was associated with pre-pregnancy obesity [41]. Recently, authors of the systematic review and meta-analysis indicated that overweight and obese women have lower chances of starting breastfeeding. The likelihood of EBF among this subgroup is also lower [42]. Our results are consistent with these findings, since chances of continuing EBF for four months among obese or overweight women with vaginal delivery were decreased. A nationwide study found that elevated BMI was widespread among Polish women at childbearing age [43]. Simultaneously, according to the European Perinatal Report, more than 10% of women in countries with pre-pregnancy BMI data were obese at the beginning of pregnancy [44]. It may suggest that decreasing rates of overweight status and obesity among women at reproductive age might positively affect peripartum period as well as breastfeeding practice.

A previous study showed that women older than 34 years were more likely to have a caesarean section [10]. However, in our study, only 30% of women delivered by caesarean section were older than 34 years of age. In our study, the odds ratio of EBF for four months was lower for younger women regardless of the mode of delivery. Women aged 18–24 were less likely to exclusively breastfeed for six months. In the study by Lande et al., the chance of EBF at four and six months was higher with an increased maternal age [45]. Other authors showed that the older maternal age of ≥ 35 and primipara were negative factors of success for EBF initiation [46]. In our study, multiparous women had a higher chance of starting EBF in the case of caesarean section. A study among Canadian women found an association between EBF for six months and factors such as living with a partner, older age at pregnancy, lower pre-pregnancy weight and being multipara. Women with earlier pregnancies might have a valuable experience in breastfeeding [47]. Previous successful breastfeeding experience and observation are positive determinants for breastfeeding [48]. Based on our findings, women who give birth for the first time, especially by caesarean section, should receive additional breastfeeding support.

The strengths of this study include a large, nationwide sample size. However, this study has some limitations. Firstly, the Computer-Assisted Web Interview might increase the possibility of receiving low quality answers. Secondly, there is a risk that self-reported weight and height could be understated or overestimated. Furthermore, inclusion criteria might be considered as a limitation of our study. There is a lack of data about women who did not start breastfeeding, since only women who initiated breastfeeding were included in this study. Moreover, this is an observational study: relationships are observed, but cause and effect conclusions cannot be drawn.

## 5. Conclusions

In conclusion, our findings confirmed that among lactating women, any breastfeeding and exclusive breastfeeding are more prevalent in the case of vaginal delivery compared to caesarean section. Overall, variables such as age, education, net income, pre-pregnancy BMI and previous pregnancies were significantly associated with exclusive breastfeeding among women who delivered by caesarean section. The results highlighted that more attention should be devoted to women who deliver by caesarean section in relation to breastfeeding. Additional breastfeeding support might be notably needed in this group. Taking relevant actions among women who deliver the baby by caesarean section might enable them to breastfeed longer. Furthermore, these findings might be valuable for healthcare professionals to develop strategies that are targeted at promoting breastfeeding in this group.

In conclusion, our results confirmed a high prevalence of caesarean section deliveries among Polish women, as well as a large decrease in the percentage of women exclusively breastfeeding during the first six months after delivery. This decrease was particularly noticeable after the fourth month following delivery. These findings highlight that breastfeeding support and promotion of exclusively breastfeeding should be directed to all women after delivery. Pregnant women and subsequently breastfeeding mothers should be under therapeutic team care.

## Figures and Tables

**Figure 1 ijerph-18-10987-f001:**
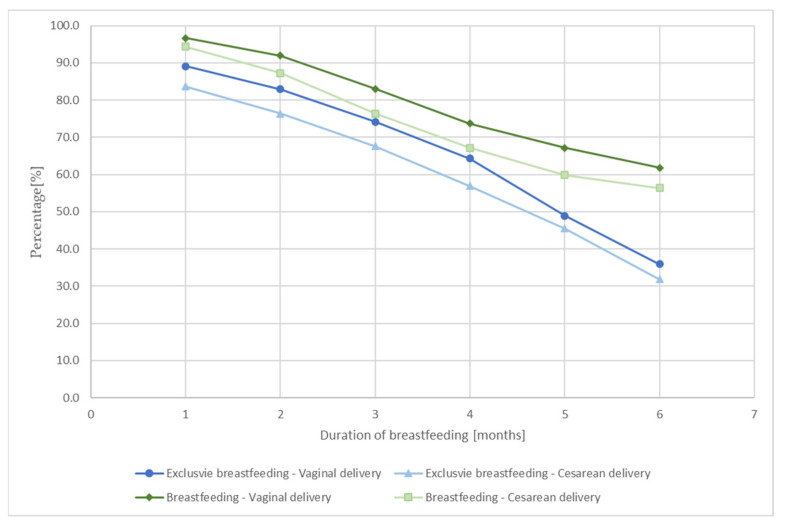
Duration of breastfeeding and exclusive breastfeeding in the case of vaginal delivery and caesarean delivery.

**Table 1 ijerph-18-10987-t001:** Demographic characteristics and gestational information in the subgroups depending on the method of delivery (*n* = 1024).

Characteristic ^1^	Method of Delivery		*p*-Value ^2^
	Vaginal *n* = 613	Caesarean *n* = 411	
Age [years]			
18–24	139 (22.7)	88 (21.4)	0.183
25–34	265 (43.2)	201 (48.9)	
≥35	209 (34.1)	122 (29.7)	
Education			
Elementary	91 (14.8)	59 (14.3)	0.105
Secondary	221 (36.0)	124 (30.2)	
Tertiary	301 (49.1)	228 (55.5)	
Place of residence			
Rural area/farm	238 (38.8)	174 (42.3)	0.481
Town ≤ 99,000 inhabitants	189 (30.8)	115 (28.0)	
City ≥ 100,000 inhabitants	186 (30.3)	122 (29.7)	
Net Income			
≤2000 PLN 3	81 (13.2)	59 (14.4)	0.795
2001–4000 PLN	181 (29.5)	122 (29.7)	
≥4001 PLN	302 (49.3)	192 (46.7)	
No answer	49 (8.0)	38 (9.2)	
Pre-pregnancy BMI [kg/m^2^]			
Underweight	31 (5.1)	26 (6.3)	0.197
Normal	346 (56.4)	208 (50.6)	
Overweight/obesity	210 (34.3)	163 (39.7)	
No data	26 (4.2)	14 (3.4)	
Primipara			
No	304 (49.6)	187 (45.5)	0.199
Pregnancy Complications			
Yes	40 (6.5)	54 (13.1)	<0.001 *

^1^ Data are given as No. (%) unless otherwise indicated. Some percentages do not total 100 because of rounding; ^2^ Chi^2^ Pearson test vaginal vs. caesarean delivery; ^3^ PLN (Polish New Zloty) = ~0.23 Euro; * statistically significant difference.

**Table 2 ijerph-18-10987-t002:** The odds ratio (OR) for exclusive breastfeeding (EBF) practice depending on the method of delivery (*n* = 1024).

Characteristic ^1^	Method of Delivery		*p*-Value ^2^
	Vaginal *n* = 613	Caesarean *n* = 411	
Starting EBF	590 (96)	380 (92)	0.008 *
OR [95% CI]	1 ref.	0.478 [0.274–0.832] *	
EBF for 4 months	394 (64)	234 (57)	0.018 *
OR [95% CI]	1 ref.	0.735 [0.569–0.949] *	
EBF for 6 months	220 (36)	131 (32)	0.185
OR [95% CI]	1 ref.	0.836 [0.641–1.089]	

^1^ Data are given as No. (%) unless otherwise indicated. Some percentages do not total 100 because of rounding; ^2^ Chi^2^ Pearson test vaginal vs. caesarean delivery; * statistically significant difference.

**Table 3 ijerph-18-10987-t003:** Factors associated with exclusive breastfeeding (EBF) practice depending on the method of delivery (*n* = 1024).

Characteristic ^1^	Starting EBF				EBF for 4 Months				EBF for 6 Months			
	Vaginal		Caesarean		Vaginal		Caesarean		Vaginal		Caesarean	
	*n* = 590	*p*-Value ^2^	*n* = 380	*p*-Value ^2^	*n* = 394	*p*-Value ^2^	*n* = 224	*p*-Value ^2^	*n* = 220	*p*-Value ^2^	*n* = 121	*p*-Value ^2^
**Age [years]**												
18–24	132 (22)	0.663	81 (21)	0.659	69 (17)	<0.001 *	35 (15)	<0.001 *	33 (15)	0.002 *	14 (11)	0.001 *
25–34	256 (43)		184 (48)		172 (44)		113 (48)		99 (45)		71 (54)	
≥35	202 (34)		115 (30)		153 (39)		86 (37)		88 (40)		46 (35)	
**Education**												
Elementary	88 (15)	0.947	56 (15)	0.034 *	44 (11)	0.001 *	19 (8)	<0.001 *	19 (9)	0.002 *	10 (8)	0.001 *
Secondary	213 (36)		120 (32)		141 (36)		71 (30)		77 (35)		31 (24)	
Tertiary	289 (49)		204 (54)		209 (53)		144 (62)		124 (56)		90 (69)	
**Place of residence**												
Rural area/farm	229 (39)	0.999	156 (41)	0.181	151 (38)	0.894	88 (38)	0.074	77 (35)	0.283	50 (38)	0.503
Town ≤ 99,000 inhabitants	182 (31)		109 (29)		124 (31)		69 (29)		75 (34)		39 (30)	
City ≥ 100,000 inhabitants	179 (30)		115 (30)		119 (30)		77 (33)		68 (31)		42 (32)	
**Net Income**												
≤2000 PLN ^3^	77 (13)	0.919	56 (15)	0.780	42 (11)	0.038 *	32 (14)	0.007 *	23 (10)	0.217	17 (13)	0.137
2001–4000 PLN	174 (29)		112 (29)		124 (31)		57 (24)		65 (29)		30 (23)	
≥4001 PLN	292 (49)		178 (47)		200 (51)		126 (54)		118 (54)		70 (54)	
No answer	47 (8)		34 (9)		28 (7)		19 (8)		14 (6)		14 (11)	
**Pre-pregnancy BMI [kg/m^2^]**												
Underweight	29 (5)	0.171	25 (7)	0.037 *	16 (4)	0.087	12 (5)	0.297	11 (5)	0.660	6 (5)	0.336
Normal	338 (57)		199 (52)		236 (60)		126 (54)		129 (59)		74 (56)	
Overweight/obesity	199 (34)		144 (38)		125 (32)		90 (38)		69 (31)		48 (37)	
No Data	24 (4)		12 (3)		17 (4)		6 (3)		11 (5)		3 (2)	
**Primipara**												0.350
No	293 (50)	0.863	181 (48)	0.002 *	198 (50)	0.660	113 (48)	0.191	119 (54)	0.095	64 (49)	
**Pregnancy Complications**												0.314
No	554 (94)	0.068	331 (87)	0.608	373 (95)	0.108	207 (88)	0.269	211 (96)	0.068	117 (89)	
**Birth weight category**												
LBW	22 (4)	0.678	32 (8)	0.657	14 (4)	0.941	16 (7)	0.270	7 (3)	0.761	10 (8)	0.858
NBW	510 (86)		301 (79)		342 (87)		190 (81)		190 (86)		105 (80)	
HBW	58 (10)		47 (12)		38 (10)		28 (12)		23 (10)		16 (12)	

^1^ Data are given as No. (%) unless otherwise indicated. Some percentages do not total 100 because of rounding; ^2^ Chi^2^ Pearson test; LBW—low birth weight; NBW—normal birth weight; HBW—high birth weight; ^3^ PLN (Polish New Zloty) = ~0.23 Euro; * statistically significant difference.

**Table 4 ijerph-18-10987-t004:** The odds ratio (OR) for exclusive breastfeeding (EBF) practice depending on the method of delivery (*n* = 1024).

Characteristic ^1^	Starting EBF OR [95% CI]		EBF for 4 Months OR [95% CI]		EBF for 6 Months OR [95% CI]	
	Vaginal *n* = 590	Caesarean *n* = 380	Vaginal *n* = 372	Caesarean *n* = 224	Vaginal *n* = 198	Caesarean *n* = 121
**Age [years]**						
18–24	0.653 [0.224–1.906]	0.704 [0.238–2.086]	0.361 [0.230–0.567] *	0.276 [0.155–0.493] *	0.428 [0.266–0.690] *	0.313 [0.159–0.616] *
25–34	0.986 [0.361–2.692]	0.659 [0.265–1.638]	0.677 [0.455–1.007]	0.538 [0.333–0.867] *	0.820 [0.566–1.188]	0.902 [0.566–1.439]
≥35	1 ref.	1 ref.	1 ref.	1 ref.	1 ref.	1 ref.
**Education**						
Elementary	1.218 [0.336–4.414]	2.196 [0.638–7.560]	0.412 [0.255–0.665] *	0.277 [0.151–0.509] *	0.377 [0.216–0.656] *	0.313 [0.151–0.649] *
Secondary	1.106 [0.444–2.752]	3.529 [1.196–10.417] *	0.776 [0.537–1.121]	0.781 [0.500–1.221]	0.763 [0.533–1.093]	0.511 [0.315–0.831] *
Tertiary	1 ref.	1 ref.	1 ref.	1 ref.	1 ref.	1 ref.
**Net Income**						
≤2000 PLN ^3^	0.688 [0.225–2.101]	1.468 [0.407–5.294]	0.549 [0.334–0.903] *	0.621 [0.343–1.123]	0.613 [0.362–1.056]	0.705 [0.374–1.332]
2001–4000 PLN	0.838 [0.306–2.291]	0.881 [0.378–2.051]	1.109 [0.748–1.645]	0.459 [0.289–0.730] *	0.874 [0.597–1.280]	0.568 [0.343–0.943] *
≥4001 PLN	1 ref.	1 ref.	1 ref.	1 ref.	1 ref.	1 ref.
**Pre-pregnancy BMI [kg/m^2^]**						
Underweight	0.343 [0.070–1.692]	1.131 [0.137–9.302]	0.064 [0.237–1.042]	0.558 [0.246–1.266]	0.925 [0.430–1.993]	0.543 [0.209–1.412]
Normal	1 ref.	1 ref.	1 ref.	1 ref.	1 ref.	1 ref.
Overweight/obesity	0.428 [0.169–1.082]	0.343 [0.151–0.779] *	0.685 [0.480–0.979] *	0.802 [0.530–1.216]	0.823 [0.574–1.181]	0.756 [0.487–1.174]
**Primipara**						
Yes	1 ref.	1 ref.	1 ref.	1 ref.	1 ref.	1 ref.
No	1.076 [0.467–2.478]	3.790 [1.520–9.448] *	1.077 [0.774–1.499]	1.300 [0.877–1.927]	1.325 [0.951–1.844]	1.219 [0.804–1.848]

^1^ Data are given as No. (%) unless otherwise indicated. Some percentages do not total 100 because of rounding; ^2^ Chi^2^ Pearson test vaginal vs. caesarean delivery; ^3^ PLN (Polish New Zloty) = ~0.23 Euro; * statistically significant difference.

## Data Availability

Not applicable.

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
