# Peer review of "The Association between the Type of Delivery and Factors Associated with Exclusive Breastfeeding Practice among Polish Women—A Cross-Sectional Study"

_ijerph, 2021, doi:10.3390/ijerph182010987_

Round 1
Reviewer 1 Report
Thank you, I have no further comments and I agree with the authors' decision.
Author Response
Response to Reviewer 1 Comments
We thank the Reviewer for their careful reading of the manuscript. We appreciate the Reviewer’s positive feedback.
Reviewer 2 Report
Thank you for your attention in addressing prior concerns about this manuscript. There are still some grammar issues that should be addressed.
line 260- the phrase "in the subgroup" is not needed
line 267 should be in the case of vaginal
line 270 "in the case of" is not needed
271- "stop exclusive breastfeeding in the beginning of lactation" -- in the beginning of lactation is a strange way to state this
274 should be the risk not a risk
the paragraph that starts on line 259-- you talk about effects of education, then of socioeconomic status then education again-- this should be cleaned up-- I agree that these are often connected but you are discussing them as separate issues so should not be intertwined
277 referring to "worse" education is really pejorative. I think it would be better to say mothers with less education
279 correlated not correlating
280 "more" education is not necessarily "better" and saying that folks with less education will not be able to comprehend the benefits of breastfeeding is again very pejorative and this section needs to be reworked.
309- i dont think that it is "chance" if a mother is breastfeeding- the wording of this is not correct
312- is there data that primiparous moms "cant start" exclusive breastfeeding- this seems unlikely and this should be reworded
348- although I certainly do think that inclusion of dieticians is a reasonable goal, I did not see any data presented in this article to support this statement.
Author Response
Response to Reviewer 2 Comments
We thank the Reviewer for their careful reading of the manuscript and their constructive remarks. We have revised the manuscript following their suggestions as is described below.
Comment 1: line 260- the phrase "in the subgroup" is not needed
Author’s response: Thank you for pointing this out. We have removed this phrase.
Comment 2: line 267 should be in the case of vaginal
Author’s response: Thank you, we have added “of”. [line 280]
Comment 3: line 270 "in the case of" is not needed
Author’s response: Thank you. We have removed phrase “in the case of”.
Comment 4: 271- "stop exclusive breastfeeding in the beginning of lactation" -- in the beginning of lactation is a strange way to state this
Author’s response: Thank you. We have reworked this sentence: In another study, women with a higher socioeconomic status were less likely to stop exclusive breastfeeding in the early postpartum period. [lines 283-285]
Comment 5: 274 should be the risk not a risk
Author’s response: Thank you. We have changed “a risk” into “the risk” [line 270]
Comment 6: the paragraph that starts on line 259-- you talk about effects of education, then of socioeconomic status then education again-- this should be cleaned up-- I agree that these are often connected but you are discussing them as separate issues so should not be intertwined
Author’s response: Thank you for pointing this out. We agree that this paragraph needed to be cleaned up so we have changed the structure of it. [lines 261-296]
Comment 7: 277 referring to "worse" education is really pejorative. I think it would be better to say mothers with less education
Author’s response: We thank the Reviewer for their suggestion on how to describe the mothers’ education in a more polite way. [line 273]
Comment 8: 279 correlated not correlating
Author’s response: Thank you. We made the change. [line 275]
Comment 9: 280 "more" education is not necessarily "better" and saying that folks with less education will not be able to comprehend the benefits of breastfeeding is again very pejorative and this section needs to be reworked.
Author’s response: Thank you for explanation why term “more education” is pejorative. We have reworked this sentence: “Parents with higher educational levels are more likely to comprehend benefits of breastfeeding”. [lines 277-278]
Comment 10: 309- i dont think that it is "chance" if a mother is breastfeeding- the wording of this is not correct
Author’s response: Thank you. We have rewritten this sentence: “Women aged 18-24 were less likely to exclusively breastfeed for six months. [line 324]
Comment 11: 312- is there data that primiparous moms "cant start" exclusive breastfeeding- this seems unlikely and this should be reworded
Author’s response: Thank you for your comment. However these data are from other author but we reworded it to make it clear: “Other authors showed that older maternal age of ≥ 35 and primipara were negative factors of success of EBF initiation”.[lines 327-328]
Comment 12: 348- although I certainly do think that inclusion of dieticians is a reasonable goal, I did not see any data presented in this article to support this statement.
Author’s response: We thank the Reviewer for their suggestion. We have decided to remove this statement as we agree that data presented in this article do not support it.
Reviewer 3 Report
Comment 1: “In conclusion, our findings confirmed that breastfeeding and exclusive breastfeeding are more prevalent among women who have vaginal delivery compared to women who have cesarean section” - As the authors state, it is very well known that cesarean births negatively impact breastfeeding - this is not a new finding.
Author’s response: We believe that although literature about breastfeeding and cesarean section is already available, further research in this area, involving different population, is necessary. Despite numerous activities aiming to encourage women to have a vaginal delivery, the proportion of cesarean sections is growing. There is a need to monitor that phenomenon.
Reviewer’s response:
Agree.
Comment 2: “Overall, variables such as age, education, net income, pre-pregnancy BMI, previous pregnancies were significantly associated with exclusive breastfeeding among women who delivered by caesarean section” - All of these factors are not modifiable and therefore are not useful when determining what and how to support women to breastfeed after caesarean.
Author’s response: Thank you for your comment. However, we maintain our opinion that many factors can affect the length of breastfeeding. Similar opinions are formulated by other authors:
− Magnano San Lio, R., Maugeri, A., La Rosa, M. C., Cianci, A., Panella, M., Giunta, G., Agodi, A., & Barchitta, M. (2021). The Impact of Socio-Demographic Factors on Breastfeeding: Findings from the "Mamma & Bambino" Cohort. Medicina (Kaunas, Lithuania), 57(2), 103. https://doi.org/10.3390/medicina57020103: Conclusions: Socio-demographic factors should be taken into account through public health strategies for improving maternal knowledge about health benefits of exclusive breastfeeding.
− Di Mattei, V. E., Carnelli, L., Bernardi, M., Jongerius, C., Brombin, C., Cugnata, F., Ogliari, A., Rinaldi, S., Candiani, M., & Sarno, L. (2016). Identification of Socio-demographic and Psychological Factors Affecting Women's Propensity to Breastfeed: An Italian Cohort. Frontiers in psychology, 7, 1872. https://doi.org/10.3389/fpsyg.2016.01872 Conclusions: It may be possible to identify women that are less inclined to breastfeed early on in pregnancy. This may aid healthcare staff to pay particular attention to women who show certain socio-demographic and psychological characteristics, so as to fulfill more focused programs.
Reviewer’s response:
Yes, it is well-known that many factors including sociodemographic characteristics influence breastfeeding duration and exclusivity.
If will be of value if programs to increase exclusive breastfeeding and the duration of breastfeeding are targeted specifically at women in pregnancy who fall into these groups.
Comment 3: “The results highlighted that more attention should devoted to women who deliver by cesarean section in relation to breastfeeding. An additional breastfeeding support might be notably needed in this group. Furthermore, these findings might be valuable for healthcare professionals” - We have known for many years that women who have had caesarean births need more breastfeeding support and this paper doesn't offer any suggestions or solutions. It just describes what it already well known in yet another population.
So much work, time, effort and money has gone into this study which does not offer one solution of how women could have been better supported or better yet have avoided a cesarean birth in the first place.
Author’s response: We would like to thank the Reviewer for the comment. This paper is part of a larger educational project titled “Nestle Healthy start to the future”. The latter project was an educational campaign for parents conducted by dieticians, aimed at supporting pregnant and breastfeeding women. This paper is part of a larger study concerning nutrition among pregnant women as well as factors influencing the duration of breastfeeding among Polish mothers. Based on this study, it was concluded that pregnant and breastfeeding women should be under therapeutic team care. In our opinion, dieticians should be part of the therapeutic team, however, within current standards, it is not the case.
Reviewer’s response:
Thank you for clarifying that this is part of a larger study aimed at influencing ‘the duration of breastfeeding among Polish mothers’, although I am not sure that dietitians are trained to provide breastfeeding support to increase breastfeeding exclusivity and duration.
I am, however, concerned that the educational project is funded by a Nestle formula manufacturer!
Formula manufacturers only profit when breastfeeding fails, so I am stunned that this has been allowed to happen.
Author Response
Response to Reviewer 3 Comments
We thank the Reviewer for careful reading of the manuscript.
Comment: Thank you for clarifying that this is part of a larger study aimed at influencing ‘the duration of breastfeeding among Polish mothers’, although I am not sure that dietitians are trained to provide breastfeeding support to increase breastfeeding exclusivity and duration.
I am, however, concerned that the educational project is funded by a Nestle formula manufacturer!
Formula manufacturers only profit when breastfeeding fails, so I am stunned that this has been allowed to happen
Author’s respond: It should be emphasized that Nestle as a grant giving organization had no influence on the interpretation of the result. Moreover, it is worth underlining that dietitians are educated on how to provide nutritional counseling for breastfeeding women. Breastfeeding women are often recommended restrictive diets by other specialist which lead to shorten breastfeeding.
- Karcz K., Lehman I., Królak-Olejnik B.: Foods to Avoid While Breastfeeding? Experiences and Opinions of Polish Mothers and Healthcare Providers. Nutrients. 2020; 12(6): 1644. doi:10.3390/ nu12061644
Reviewer 4 Report
This manuscript reports the findings of a large study of Polish women, whose breast feeding success was determined retrospectively. The data are valuable. The only item that is missing is some estimate of how close the study sample came to being representative of all Polish women.
- Line 49: change "month" to "months"
- Study groups: We need to be told how the study women were selected. And we need to know how close the sample came to being representative of the population.
- Line 163: add "significant" before "differences"
- Table 3: there are 6 columns of p-values and it is not clear to whom the values in columns 2, 4 and 6 pertain
- Line 196: insert "a" before "50%"
- Line 229: change "cesarian" to "cesarean" and multiple other places
- Line 275: change "doubled risk" to "double the risk"
- Line 279: delete "In"
- Line 299: change "month" to "months"
Author Response
Response to Reviewer 4 Comments
Comment 1: Line 49: change "month" to "months"
Author’s response: Thank you. We changed “month” to “months” [line 50]
Comment 2: Study groups: We need to be told how the study women were selected. And we need to know how close the sample came to being representative of the population.
Author’s response: We agree that information on how the study women were selected is necessary. In section Materials and Methods we described the “Study Participants and Collecting Data”. We have added the information that the external research company provided the sampling frame and the IT tools needed for the field work as below:
Eligibility criteria were: being an adult (≥18 years old), breastfeeding woman and being a mother of an infant or a toddler aged 6-18 months. Exclusion criteria included being pregnant, being a woman who has never breastfed, being the mother of an infant aged <6 months or a toddler aged over 18 months.
The study was conducted using the Computer-Assisted Web Interview (CAWI). The tool used to carry out the research was a questionnaire. Respondents were asked to complete the individual electronic form on the web panel (Opinie.pl) belonging to the independent, external research Agency IQS. This web panel has existed since 2007. Currently, it has 110.000 active users including 10.500 mothers of children born between 2015-2018. Participation in the study was anonymous and voluntary and women were not compensated for the participation. The research company provided the sampling frame and the IT tools needed for the field work. The time for filling out the questionnaire did not exceed 15 minutes
Comment 3: Line 163: add "significant" before "differences"
Author’s response: Thank you for that suggestion, however it had been already added.
Comment 4: Table 3: there are 6 columns of p-values and it is not clear to whom the values in columns 2, 4 and 6 pertain
Author’s response: Thank you for pointing this out. We have changed layout of upper row “characteristic” to make it more clear.
Comment 5: Line 196: insert "a" before "50%"
Author’s response: Thank you for pointing this out. We have inserted “a”. [line 205]
Comment 6: Line 229: change "cesarian" to "cesarean" and multiple other places
Author’s response: Thank you for the reminder. We checked spelling in other places and changed all “cesarian” to “cesarean”.
Comment 7: Line 275: change "doubled risk" to "double the risk"
Author’s response: Thank you we have changed this phrase. [line 299]
Comment 8: Line 279: delete "In"
Author’s response: Thank you for the reminder. We have deleted that.
Comment 9: Line 299: change "month" to "months"
Author’s response: Thank you we have changed month to months. [line 325]
This manuscript is a resubmission of an earlier submission. The following is a list of the peer review reports and author responses from that submission.
Round 1
Reviewer 1 Report
In conclusion, our findings confirmed that breastfeeding and exclusive breastfeeding are more prevalent among women who have vaginal delivery compared to women who have cesarean section.
As the authors state, it is very well known that cesarean births negatively impact breastfeeding - this is not a new finding.
Overall, variables such as age, education, net income, pre-pregnancy BMI, previous pregnancies were significantly associated with exclusive breastfeeding among women who delivered by caesarean section.
All of these factors are not modifiable and therefore are not useful when determining what and how to support women to breastfeed after caesarean.
The results highlighted that more attention should devoted to women who deliver by cesarean section in relation to breastfeeding. An additional breastfeeding support might be notably needed in this group. Furthermore, these findings might be valuable for healthcare professionals.
We have known for many years that women who have had caesarean births need more breastfeeding support and this paper doesn't offer any suggestions or solutions. It just describes what it already well known in yet another population.
So much work, time, effort and money has gone into this study which does not offer one solution of how women could have been better supported or better yet have avoided a cesarean birth in the first place.
This paper should not be published in a special issue of your journal about 'protecting, promoting and supporting breastfeeding'.
It does none of these...
Author Response
We thank the Reviewer for his careful revision of the manuscript and his constructive remarks. Please find below a detailed point-by-point response to all comments.
Comment 1: “In conclusion, our findings confirmed that breastfeeding and exclusive breastfeeding are more prevalent among women who have vaginal delivery compared to women who have cesarean section” - As the authors state, it is very well known that cesarean births negatively impact breastfeeding - this is not a new finding.
Author’s response: We believe that although literature about breastfeeding and cesarean section is already available, further research in this area, involving different population, is necessary. Despite numerous activities aiming to encourage women to have a vaginal delivery, the proportion of cesarean sections is growing. There is a need to monitor that phenomenon.
Comment 2: “Overall, variables such as age, education, net income, pre-pregnancy BMI, previous pregnancies were significantly associated with exclusive breastfeeding among women who delivered by caesarean section” - All of these factors are not modifiable and therefore are not useful when determining what and how to support women to breastfeed after caesarean.
Author’s response: Thank you for your comment. However, we maintain our opinion that many factors can affect the length of breastfeeding. Similar opinions are formulated by other authors:
− Magnano San Lio, R., Maugeri, A., La Rosa, M. C., Cianci, A., Panella, M., Giunta, G., Agodi, A., & Barchitta, M. (2021). The Impact of Socio-Demographic Factors on Breastfeeding: Findings from the "Mamma & Bambino" Cohort. Medicina (Kaunas, Lithuania), 57(2), 103. https://doi.org/10.3390/medicina57020103: Conclusions: Socio-demographic factors should be taken into account through public health strategies for improving maternal knowledge about health benefits of exclusive breastfeeding.
− Di Mattei, V. E., Carnelli, L., Bernardi, M., Jongerius, C., Brombin, C., Cugnata, F., Ogliari, A., Rinaldi, S., Candiani, M., & Sarno, L. (2016). Identification of Socio-demographic and Psychological Factors Affecting Women's Propensity to Breastfeed: An Italian Cohort. Frontiers in psychology, 7, 1872. https://doi.org/10.3389/fpsyg.2016.01872 Conclusions: It may be possible to identify women that are less inclined to breastfeed early on in pregnancy. This may aid healthcare staff to pay particular attention to women who show certain socio-demographic and psychological characteristics, so as to fulfill more focused programs.
Comment 3: “The results highlighted that more attention should devoted to women who deliver by cesarean section in relation to breastfeeding. An additional breastfeeding support might be notably needed in this group. Furthermore, these findings might be valuable for healthcare professionals” - We have known for many years that women who have had caesarean births need more breastfeeding support and this paper doesn't offer any suggestions or solutions. It just describes what it already well known in yet another population.
So much work, time, effort and money has gone into this study which does not offer one solution of how women could have been better supported or better yet have avoided a cesarean birth in the first place.
Author’s response: We would like to thank the Reviewer for the comment. This paper is part of a larger educational project titled “Nestle Healthy start to the future”. The latter project was an educational campaign for parents conducted by dieticians, aimed at supporting pregnant and breastfeeding women. This paper is part of a larger study concerning nutrition among pregnant women as well as factors influencing the duration of breastfeeding among Polish mothers. Based on this study, it was concluded that pregnant and breastfeeding women should be under therapeutic team care. In our opinion, dieticians should be part of the therapeutic team, however, within current standards, it is not the case.
Reviewer 2 Report
This paper reports on an online survey of Polish mothers who breastfed for at least some portion of their child's first six months. The authors are looking to identify factors that are associated with exclusive breastfeeding at birth, 2, 4, 6 months. Identifying factors that can assist in creating effective interventions to support breastfeeding is an important goal but it was not clear how much this paper will add to this knowledge.
Since all of the women in the study reported some component of breastfeeding, it is hard to know anything about the actual prevalence of breastfeeding in these various subcategories as a large number of women are already excluded. It was also not clear if women were paid to respond to this survey, and thus if there was some social desirability that affected their response and identification as eligible for the study.
I feel that there are significant limitations to the study that need to be addressed in the write up.
In the introduction, it would be useful to discuss what is known about breastfeeding and the various factors you have identified as being of interest to understand why you have selected them-- instead it was only about c/s and breastfeeding- but since this study doesnt actually look at breastfeeding and c/s- this wasnt as useful. There are some errors in the introduction. On line 34 you say that reference 4 discusses COVID19 and breastfeeding. However this citation was published in June of 2019 so there is no way that it talks about covid19 infection in folks who were breastfed in the past.
line 38- us should be as
line 43 "consentaneously" is not a word
line 55 "more attention should be paid to early breastfeeding after c/s"- this seems like a line that should be in your conclusion than in the introduction - otherwise it seems like your information is already available.
line 61/62- this sentence is awkward and should be reworked
line 63 the sentence that starts "Therefore" is not a clear transition. You say that c/s rates are higher in the prior paragraph and then say that other factors should be taken into account in the perinatal period- but you have not introduced those other factors.
line 64- again you say that women with c/s need additional interventions- which should be the conclusion of your study not the introduction
line 72 also awkwardly worded
line 76- what is IQS?
line 175 respondants needs an apostrophe: respondants'
line 215 this sentence is awkward as it implies the women were in labor for 6 months- needs punctuation or just reworking
line 247 while i agree that women should be told about the risks and benefits of vaginal delivery vs c/s, this was not discussed at all in the paper and this sentence doesnt fit
line 251- what "special needs" related to c/s? (I can imagine what you are talking about but this didnt really come from anywhere
line 274 saying that folks who have lower education will not be able to comprehend the benefits of breastfeeding came across as derogatory to me- may wish to reword. I would imagine job type, family support, economics etc-- are all things that may correlate with education and not with ability to comprehend
line 281 us should be as
line 298 women who are >34 are more likely to have c/s- not sure how this correlates with the odds ratio of breastfeeding in younger women.
line 322 Without understanding anything about women who were not breastfeeding, you really cannot make any statement about the prevalence of breastfeeding in these two populations as you have no denominator.
line 327 attention should devoted" needs a "be" before devoted
line 328 is basically a repeat of the prior sentence and should be deleted.
Author Response
We thank the Reviewer for his careful revision of the manuscript and his constructive remarks. Please find below a detailed point-by-point response to all comments.
Comment 1: This paper reports on an online survey of Polish mothers who breastfed for at least some portion of their child's first six months. The authors are looking to identify factors that are associated with exclusive breastfeeding at birth, 2, 4, 6 months. Identifying factors that can assist in creating effective interventions to support breastfeeding is an important goal but it was not clear how much this paper will add to this knowledge.
Since all of the women in the study reported some component of breastfeeding, it is hard to know anything about the actual prevalence of breastfeeding in these various subcategories as a large number of women are already excluded. It was also not clear if women were paid to respond to this survey, and thus if there was some social desirability that affected their response and identification as eligible for the study.
Author’s response: Thank you for the constructive review. The Reviewer’s comments have improved our manuscript. We highlighted changes which addressed the Reviewer’s comments and suggestions in yellow. This paper is part of a larger study concerning factors influencing the duration of breastfeeding among Polish mothers. Nonprobability sampling was used in this study and the inclusion criteria were: being a breastfeeding mother at the time of the study or a mother who has been breastfeeding in the past. The aim of the whole project was to analyze various factors that influenced the length of breastfeeding. In this article, we focused only on the type of delivery. We have specified this limitation in the discussion section: “Furthermore, inclusion criteria might be considered as a limitation of our study. There is lack of data about women who did not start breastfeeding since only women who initiated breastfeeding were included in this study”.
Women were not paid to respond to the survey. We added this following statement: “Participation in the study was voluntary and women were not compensated for the participation.” [lines 91-93]
Comment 2: I feel that there are significant limitations to the study that need to be addressed in the write up.
Author’s response: We agree that there are limitations to this study, however we addressed them in the manuscript, as follows: “However, this study has some limitations. Firstly, Computer-Assisted Web Interview might increase the possibility to get low quality answers. Secondly, there is a risk that self-reported weight and height could be understated or overestimated. Furthermore, inclusion criteria might be considered as a limitation of our study. There is lack of data about women who did not start breastfeeding since only women who initiated breastfeeding were included in this study. Moreover, this is an observational study which allows to observe relationships but cause and effect conclusions cannot be drawn.” [lines 320-327]
Comment 3: In the introduction, it would be useful to discuss what is known about breastfeeding and the various factors you have identified as being of interest to understand why you have selected them-- instead it was only about c/s and breastfeeding- but since this study doesnt actually look at breastfeeding and c/s- this wasnt as useful. There are some errors in the introduction. On line 34 you say that reference discusses COVID19 and breastfeeding. However this citation was published in June of 2019 so there is no way that it talks about covid19 infection in folks who were breastfed in the past.
Author’s response: Thank you for this suggestion. It would have been interesting to explore this aspect. However, the introduction is based on available literature and we aimed at presenting general information about breastfeeding benefits. This paper is part of a larger study concerning factors influencing the duration of breastfeeding among Polish mothers. Sociodemographic factors were related to the comparison between two groups which were distinguished based on type of delivery.
Thank you for pointing out the error in reference 4. We made a mistake in the citation – it should be the following article: Didikoglu A, Maharani A, Pendleton N, Canal MM, Payton A. Early life factors and COVID-19 infection in England: A prospective analysis of UK Biobank participants. Early Hum Dev. 2021;155:105326. doi:10.1016/j.earlhumdev.2021.105326. We reconsidered whether we should use this reference and we have decided to remove it. [line 33]
Comment 4: line 38- us should be as
Author’s response: Thank you. We made the change. [line 37]
Comment 5: line 43 "consentaneously" is not a word
Author’s response: Thank you for pointing this out, we have removed the word “consentaneously” [line 42]
Comment 6: line 55 "more attention should be paid to early breastfeeding after c/s"- this seems like a line that should be in your conclusion than in the introduction - otherwise it seems like your information is already available.
Author’s response: Thank you for this suggestion, we agree that this sentence is unnecessary in this place. We have deleted it. [line 54]
Comment 7: line 61/62- this sentence is awkward and should be reworked
Author’s response: Thank you for the reminder. We rewrote this sentence. [line 61]
Comment 8: line 63 the sentence that starts "Therefore" is not a clear transition. You say that c/s rates are higher in the prior paragraph and then say that other factors should be taken into account in the perinatal period- but you have not introduced those other factors
Author’s response: Thank you. We deleted “therefore” as it was not the suitable word. [line 64]
Comment 9: line 64- again you say that women with c/s need additional interventions- which should be the conclusion of your study not the introduction
Author’s response: Thank you for this suggestion. We delated this sentence for introduction and reworked conclusions a bit.
Comment 10: line 72 also awkwardly worded
Author’s response: Thank you. We have simplified the sentence. [line 73]
Comment 11: line 76- what is IQS? Author’s response: IQS Agency is a Polish research-analytics company – we have added this description to the manuscript. [line 77]
Comment 12: line 175 respondants needs an apostrophe: respondants'
Author’s response: Thank you for the reminder. We have added an apostrophe. [line 177]
Comment 13: line 215 this sentence is awkward as it implies the women were in labor for 6 months- needs punctuation or just reworking
Author’s response: We thank the Reviewer for pointing this out. We have reworked this sentence. [line 219]
comment 14: line 247 while i agree that women should be told about the risks and benefits of vaginal delivery vs c/s, this was not discussed at all in the paper and this sentence doesnt fit
Author’s response: Thank you for pointing this out. We agree with this comment and we decided to remove this sentence. [line 250]
Comment 15: line 251- what "special needs" related to c/s? (I can imagine what you are talking about but this didnt really come from anywhere
Author’s response: Thank you for this comment. We have explained what kind of breastfeeding support mothers who deliver by cesarean section should get. [line 253-257]
Comment 16: line 274 saying that folks who have lower education will not be able to comprehend the benefits of breastfeeding came across as derogatory to me- may wish to reword. I would imagine job type, family support, economics etc-- are all things that may correlate with education and not with ability to comprehend
Author’s response: Thank you for pointing this out. We revised this sentence. [line 276]
Comment 17: line 281 us should be as
Author’s response: Thank you. We made the change. [line 286]
Comment 18: line 298 women who are >34 are more likely to have c/s- not sure how this correlates with the odds ratio of breastfeeding in younger women.
Author’s response: Thank you for your comment. We agree that this part was not clear enough. We added a sentence which was missing and which correlates with the previous one. [lines 304-306]
Comment 19: line 322 Without understanding anything about women who were not breastfeeding, you really cannot make any statement about the prevalence of breastfeeding in these two populations as you have no denominator.
Author’s response: Thank you very much for your comment. We have reworked conclusions
Comment 20: line 327 attention should devoted" needs a "be" before devoted
Author’s response: Thank you. We have decided to rewrite conclusions so these phrase was removed.
Comment 21: line 328 is basically a repeat of the prior sentence and should be deleted.
Author’s response: Thank you for pointing this out. We agree and have removed this sentence. [line 331-333]
Reviewer 3 Report
I would like to thank the editor for the opportunity to review the manuscript entitled "The association between the type of delivery and exclusive breastfeeding practices among Polish women - a cross-sectional study". I read the paper with great interest and I congratulate with the authors for the well-conducted study and well-written manuscript.
I have only a minor suggestion: in the introduction, they state that being breastfed can be associated with the risk of COVID-19, giving as reference "Zhang et al." (number 4). I am not sure if it is the correct reference; nevertheless, I feel that it might be imprudent to give such statements, as undoubtedly more studies are needed and during these times it is worth to be more cautious about the information we give on COVID-19, after too many fairly daring comments that led to a sort of insecureness and confusion of the general population.
Overall a good paper, congratulations.
Author Response
We thank the Reviewer for his careful revision of the manuscript and his constructive remarks. Please find below a detailed point-by-point response to all comments.
Comment 1: I would like to thank the editor for the opportunity to review the manuscript entitled "The association between the type of delivery and exclusive breastfeeding practices among Polish women - a cross-sectional study". I read the paper with great interest and I congratulate with the authors for the well-conducted study and well-written manuscript.
Author’s response: We would like to thank the Reviewer for the comment and for the classification of the manuscript as well-written.
Comment 2: I have only a minor suggestion: in the introduction, they state that being breastfed can be associated with the risk of COVID-19, giving as reference "Zhang et al." (number 4). I am not sure if it is the correct reference; nevertheless, I feel that it might be imprudent to give such statements, as undoubtedly more studies are needed and during these times it is worth to be more cautious about the information we give on COVID-19, after too many fairly daring comments that led to a sort of insecureness and confusion of the general population.
Author’s response: Thank you for pointing this out. We made a mistake in the citation – it should be the following article: Didikoglu A, Maharani A, Pendleton N, Canal MM, Payton A. Early life factors and COVID-19 infection in England: A prospective analysis of UK Biobank participants. Early Hum Dev. 2021;155:105326. doi:10.1016/j.earlhumdev.2021.105326. We agree with the Reviewer’s suggestion that more studies are needed in this area and more caution is needed while publishing the information about COVID-19. This reference has been deleted.